# Leveraging Deep Learning for Time-Series Extrinsic Regression in Predicting the Photometric Metallicity of Fundamental-Mode RR Lyrae Stars

**DOI:** 10.3390/s24165203

**Published:** 2024-08-11

**Authors:** Lorenzo Monti, Tatiana Muraveva, Gisella Clementini, Alessia Garofalo

**Affiliations:** INAF—Osservatorio di Astrofisica e Scienza dello Spazio di Bologna, via Piero Gobetti 93/3, 40129 Bologna, Italy; tatiana.muraveva@inaf.it (T.M.); gisella.clementini@inaf.it (G.C.); alessia.garofalo@inaf.it (A.G.)

**Keywords:** deep learning, recurrent neural networks, convolutional neural networks, time-series extrinsic regression

## Abstract

Astronomy is entering an unprecedented era of big-data science, driven by missions like the ESA’s Gaia telescope, which aims to map the Milky Way in three dimensions. Gaia’s vast dataset presents a monumental challenge for traditional analysis methods. The sheer scale of this data exceeds the capabilities of manual exploration, necessitating the utilization of advanced computational techniques. In response to this challenge, we developed a novel approach leveraging deep learning to estimate the metallicity of fundamental mode (ab-type) RR Lyrae stars from their light curves in the Gaia optical G-band. Our study explores applying deep-learning techniques, particularly advanced neural-network architectures, in predicting photometric metallicity from time-series data. Our deep-learning models demonstrated notable predictive performance, with a low mean absolute error (MAE) of 0.0565, the root mean square error (RMSE) of 0.0765, and a high $R^{2}$ regression performance of 0.9401, measured by cross-validation. The weighted mean absolute error (wMAE) is 0.0563, while the weighted root mean square error (wRMSE) is 0.0763. These results showcase the effectiveness of our approach in accurately estimating metallicity values. Our work underscores the importance of deep learning in astronomical research, particularly with large datasets from missions like Gaia. By harnessing the power of deep-learning methods, we can provide precision in analyzing vast datasets, contributing to more precise and comprehensive insights into complex astronomical phenomena.

## 1. Introduction

RR Lyrae stars are low-mass, core-helium-burning, radially pulsating variable stars representing an old stellar population (for completeness of information, see Smith, H.A. (1995) [1]). RR Lyrae stars are unique objects because their intrinsic properties, such as distance and metallicity ([Fe/H]), can be determined from easily observed photometric parameters (e.g., apparent magnitude, pulsation period, etc.). These characteristics have made them valuable distance indicators and metallicity tracers in astrophysical studies of the Local Group galaxies (Tanakul & Sarajedini, 2018; Clementini et al., 2019; Dékány & Grebel, 2020; Bhardwaj, 2022) [2,3,4,5]. RR Lyrae stars are classified into three types: fundamental mode (ab type), which have asymmetric light curves with longer periods; first-overtone (c type) RR Lyrae stars, characterized by symmetric, sinusoidal light curves and shorter periods; and double-mode pulsation (d type) RR Lyrae stars, which exhibit simultaneous pulsation in both the fundamental and first overtone modes, resulting in complex light curves (Smith, H.A (1995) [1]).

The empirical relationship between the shape of the fundamental mode (ab type) RR Lyrae stars’ light curve and its metallicity has been recognized since the work of Jurcsik and Kovács (1996) [6]. However, accurately calibrating this relationship across multiple photometric bands has presented challenges, primarily due to a limited amount of stars with high-dispersion spectroscopic (HDS) measurements of metallicity. Various empirical formulae have emerged over the past 25 years to predict metallicity from the Fourier regression parameters of light curves, using different methodologies. Given the scarcity of HDS metallicities, most approaches rely on low-dispersion spectroscopic or spectrophotometric estimates or transfer predictive formulae between different passbands, introducing inherent noise and metallicity dependency (Layden, 1994; Smolec, 2005; Ngeow et al., 2016; Skowron et al., 2016; Mullen et al., 2021) [7,8,9,10,11]. The limited, heterogeneous data and errors in regressors often lead to systematic biases in metallicity prediction formulae. Recently, the number of RR Lyrae stars with HDS metallicity estimates has increased (Crestani et al., 2021; Gilligan et al., 2021) [12,13], prompting a revision of earlier photometric metallicity estimation methods.

In this sense, deep-learning models have emerged as powerful tools for various astronomical tasks, including metallicity estimation of RR Lyrae stars. These models offer the potential to capture complex relationships between light curve features and metallicity, thus improving prediction accuracy. However, the effectiveness of deep-learning models hinges on the availability of large, high-quality datasets. In the context of RR Lyrae stars, obtaining such datasets can be challenging due to the limited number of stars with precise spectroscopic measurements. Additionally, the heterogeneity and inherent noise in observational data pose significant challenges for model training and generalization. Despite these challenges, the potential benefits of deep-learning models for the metallicity estimation of RR Lyrae stars are substantial.

The era of the Gaia mission has completely revolutionized the quality and quantity of data at our disposal for more than 1.8 billion sources, among them, RR Lyrae stars. With the most recent Gaia Data Release (DR3, Vallenari et al., 2023) [14], almost 271 thousand RR Lyrae stars distributed all over the sky have been released up to optical G band ∼21 mag (Clementini et al., 2023) [15]. For almost all of them, Gaia DR3 provided, together with astrometric and photometric products, pulsation properties (period, amplitude, Fourier parameters, etc.) and photometric time-series in Gaia bands.

By exploiting the large catalogues of RR Lyrae stars, such as those provided by the ESA Gaia mission (Clementini et al., 2023) [15], and advanced neural-network architectures, these models have the capacity to improve the accuracy and precision of metallicity predictions, ultimately enhancing our understanding of stellar populations traced by RR Lyrae and the evolution of galaxies in which they are located. Using photometric metallicity estimates, we established new empirical predictive models for the metallicity of fundamental-mode (RRab) RR Lyrae stars from their light curves. With a sufficiently large and accurate training set, direct regression to light curves data is feasible, eliminating the customary intermediate step of calibrating spectral indices, such as the Fourier parameters of the light curves, to HDS metallicity values.

## 2. Background and Related Works

Recent advancements in deep learning have significantly impacted the field of astronomy and the analysis of time-series data related to variable stars. In particular, it is possible to divide this research area into three categories as follows.

### 2.1. Time-Series Classification

Brett Naul and colleagues (2018) [16] presented an innovative recurrent neural-network (RNN) approach that uses unsupervised auto-encoding for the efficient and accurate classification of such variable stars. This methodology demonstrated competitive performance compared to other state-of-the-art methods, highlighting the effectiveness of deep-learning techniques in dealing with the irregular time-series data commonly encountered in astronomical observations. Moreover, in a pivotal study by Aguirre, Pichara, and Becker (2019) [17], a novel deep-learning model integrating convolutional units was developed, designed to classify variable stars from their light curves. This study showcases the potential of this method to revolutionize the process of categorizing vast arrays of celestial phenomena across multi-survey data. A study by Jamal and Bloom (2020) [18] compared various neural-network architectures, including RNN, dilated temporal convolutional neural networks (dTCNs), long-short term memory neural networks, gated recurrent units, and temporal convolutional neural networks (tCNNs), for astronomical time-series classification. This study highlighted the importance of choosing the right architecture for specific astronomical tasks. Kang et al. (2023) [19] addressed the challenge of data imbalance in classifying periodic variable stars by proposing an ensemble augmentation method combining RNN and CNN for improved accuracy. This was a significant step forward in enhancing classification accuracy despite dataset limitations. Furthermore, Allam, Peloton, and McEwen (2023) [20] demonstrated that deep compression methods could achieve a significant reduction in model size while maintaining classification performance. This improvement in model size enhances inference latency and throughput for time-critical events in real-time astronomical data analysis.

This indicates the potential of deep learning in enhancing the real-time analyses of astronomical events. These studies illustrate the diverse applications and benefits of deep learning in the field of astronomy in the analysis and classification of time-series data related to variable stars. From improving classification accuracy to enhancing real-time event analysis, deep learning continues to push the boundaries of what is possible in astronomical research.

### 2.2. Time-Series Clustering

Deep learning and machine learning have continually emerged as powerful tools in astronomical research in the analysis of the time-series data of variable stars. While some studies focus on using traditional machine-learning and deep-learning approaches for stellar classification and analysis, there is growing interest in leveraging deep learning for clustering and analyzing variable stars within astronomical time-series data. A study conducted by Rebbapragada et al. (2009) [21] focused on an unsupervised anomaly detection approach tailored for extensive collections of unsynchronized periodic time-series data. It generates a prioritized catalog of both overarching and localized anomalies. The method computes anomaly scores for each light curve concerning a cluster of centroids derived from a modified k-means clustering algorithm. Thanks to the work done by researchers, it is possible to obtain scalability for large datasets by employing sampling techniques. The performance has been validated on various datasets, including light-curve data, showcasing its proficiency in identifying known anomalies. Notably, David J. Armstrong et al. (2015) [22] implemented innovative methodologies, including Kohonen self-organizing maps (SOMs) and random forest (RF) techniques, for classifying variable stars in the K2 mission fields. Further, the study from Mackenzie, Pichara, and Protopapas (2016) [23] focuses on the automatic classification of variable stars using clustering algorithms, which significantly hinges on how light curves are represented. The research introduces a novel feature learning algorithm tailored for variable objects. It initiates by extracting numerous light curve subsequences, followed by clustering these to discover common local patterns in the time series. Subsequently, it utilizes representatives of these common patterns to transform the light curves into a new representation. This new data representation is then employed to train a classifier. The novelty of the approach lies in its ability to learn features from both labeled and unlabeled light curves, thereby mitigating the bias inherent in using only labeled data for feature learning. Moreover, the approach illustrated by Valenzuela and Pichara (2018) [24] introduces an unsupervised learning approach for classifying variable stars, leveraging the similarity among light curves to categorize them. This represents a significant advancement in the field of astronomy, particularly in the automated classification of variable stars, by relying on the intrinsic patterns present in the light curves without the need for labeled data. The methodology proposed in Valenzuela’s work aligns with the broader efforts in astronomy to employ machine-learning techniques for efficient data analysis given the massive volumes of data generated by astronomical surveys. For example, the use of unsupervised learning algorithms to classify the chemistry of long-period variable stars based on BP and RP spectra from Gaia Third Release (DR3) shows the versatility of unsupervised methods in identifying complex astronomical phenomena, as described by Sanders et al. (2023) [25].

These studies exemplify the evolving landscape of astronomical research, where both machine learning and deep learning are employed to tackle the intricate task of analyzing and classifying variable stars through time-series data. The application of these technologies opens new avenues for exploring and understanding the dynamic behaviors of ‘resolved’ stars within our galaxy and beyond.

### 2.3. Time-Series Regression

The utilization of deep learning in astronomy, particularly for time-series regression involving variable stars, presents promising advancements across various facets of stellar study. Surana et al., (2021) [26] explored the prediction of crucial star formation properties like stellar mass, star formation rate, and dust luminosity by leveraging deep-learning techniques as an alternative to traditional stellar population synthesis models. In the chase for uncovering variable stars, Noughani and Kotulla (2020) [27] harnessed a decade-long dataset from the Sloan Digital Sky Survey telescope to identify variable stars, accurately estimate their variability periods, and elucidate the shape of their brightness fluctuations over time. This work is integral to propelling forward the field of time-domain astrophysics. RNNs have also found applications in the astronomical domain, as showcased by Flores et al. (2022) [28], who utilized RNNs to estimate the physical parameters of O-type stars (hot, blue-white stars of spectral type O) in the optical region of stellar spectra, exemplifying the versatility of deep-learning models in stellar parameter estimation. In a groundbreaking study by Dékány and Grebel (2022) [29], deep learning, specifically the deployment of long short-term memory recurrent neural networks, was applied to the task of regressing the metallicity abundance from time-series light curves observed in Gaia’s optical G and near-infrared VISTA $K_{s}$ bands. This methodology allowed for the processing of serialized data inherent to light curves, resulting in low mean absolute errors and high regression performance, making it a significant advancement in the metallicity estimation of RR Lyrae stars based on their photometric observations. Complementing this work, a study by Dékány, Grebel, and Pojmański (2021) [30] further explored the metallicity prediction capabilities through machine learning. By employing a combination of machine-learning methods alongside Bayesian regression, they established empirical relationships between metallicity abundance and various light-curve parameters for RRab and first-overtone (RRc) stars. This approach achieved mean absolute prediction errors of 0.16 dex and 0.18 dex, respectively, demonstrating the precision possible when combining statistical models with analysis of stellar light curves.

Collectively, these studies manifest the transformative impact of deep learning on the astrophysics field concerning time-series regression tasks for variable stars. By enabling more nuanced analysis, improved accuracy in parameter estimation, and metallicity estimation tasks, deep-learning tools are proving essential in pushing the boundaries of our astronomical understanding and capabilities.

## 3. Photometric Data and Data Preprocessing

Building on the preceding section, our objective aligns with the domain known as Section 2.3. Specifically, we focus on a variant termed time-series extrinsic regression (TSER), where the goal is to learn the relationship between time-series data (photometric light curves) and a continuous scalar value (metallicity value referring to the stellar object).

To delve deeper, as outlined in the paper by Tan et al. (2021) [31], this task relies on the entirety of the series rather than emphasizing recent over past values, as seen in *time-series forecasting (TSF)*. The contrast between *time-series classification (TSC)* and TSER lies in TSC’s mapping of a time series to a finite set of discrete labels, whereas TSER predicts a continuous value from the time series. Therefore, formally speaking, the definition of TSER mirrors the one elucidated in the aforementioned paper:

A TSER model is represented as a function T → R, where T denotes a set of time series. The objective of time-series extrinsic regression is to derive a regression model from a dataset, $D=\left\{\left(t_{1},r_{1}\right),...,\left(t_{n},r_{n}\right)\right\}$, where each $t_{i}$ represents a time series and $r_{i}$ represents a continuous scalar value.

Keeping this in consideration, our initial step involves retrieving the dataset from the Gaia Data Release 3 (DR3) catalogue of RR Lyrae stars (Clementini et al., 2023) [15], for which we have photometric metallicity estimates (Muraveva et al., 2024 [32]). Specifically, we have applied the following selection criteria:

σ[Fe/H]≤0.4 dex; AmpG≤1.4 mag; $N_{ep}>=50;$ $\sigma \phi_{31}\leq 0.10.$

Here, σ[Fe/H] represents the uncertainty associated with the metallicity value, AmpG denotes the peak-to-peak amplitude of the light curves, $N_{ep}$ signifies the number of epochs in the G-band, and $\sigma \phi_{31}$ is the parameter of the Fourier decomposition of the light curve. Additionally, it is verified that the phase is less than 1.0 and that the period values are not missing. Other data values are ensured to be complete, as they are sourced from the Gaia DR3 catalog. Our final sample consists of 6002 RRab stars, with 4801 samples used for the training set to develop and refine our models, and 1201 sample assigned to the validation set to evaluate and validate the model’s performance. An illustrative example of the photometric dataset adopted in this study is provided in Table 1 below.

Therefore, we have applied a fundamental step in the analysis of variable stars, which is essential for extracting meaningful information from observational data. Phase folding and alignment are techniques commonly used in the study of variable stars, particularly those with periodic variability such as pulsating stars:*Phase Folding*: In phase folding, observations of a variable star’s brightness over time are transformed into a phase-folded light curve. This involves folding the observations based on the star’s known or estimated period. The period is the duration of one complete cycle of variability, such as the time it takes for a star to pulsate or undergo other periodic changes. By folding the observations, multiple cycles of variability are aligned so that they overlap, simplifying the analysis of the star’s variability pattern. This technique allows astronomers to better understand the periodic behavior of variable stars and to compare observations more effectively.*Phase Alignment*: Alignment refers to the process of adjusting or aligning multiple observations of a variable star’s light curve to a common reference point. This is particularly important when studying stars with irregular or asymmetric variability patterns. By aligning observations, astronomers can more accurately compare the shape, timing, and amplitude of variations in the star’s brightness. This helps in identifying patterns, detecting periodicity, and studying the underlying physical mechanisms driving the variability. Therefore, RRab-type stars have a sawtooth-shaped light curve, which is indeed asymmetric, with a rapid rise and a slow decline. For these types of star, the phase alignment mentioned is particularly important.

By applying the well-known formula in variable stars literature to all light curves within the previously selected catalog:(1)$$phase=\left(\frac{T-Epoch_{max}}{P}\right)-\;\mathrm{mod}\;\left(\frac{T-Epoch_{max}}{P}\right)$$
where *T* represents the observation time and $Epoch_{max}$ denotes the epoch at the maximum light of the source during the pulsation cycle that is expressed in Barycentric Julian Day (BJD) for Gaia’s sources, and *P* is a pulsation period. As shown in Figure 1, following the application of the aforementioned method, our dataset for training deep-learning models is composed of 6002 G-band light curves of RR Lyrae stars.

Once the data have been phase-folded, a method called *smoothing spline* has been applied, using the SciPy library ( https://scipy.org/, accessed on 4 July 2024) version 1.10.1, to minimize fluctuations, noise, and outliers, and to obtain the same number of points for each light curve.

More in-depth, *smoothing spline* is a method used for fitting a smooth curve to a set of data points effectively balancing between accurately representing the data and minimizing fluctuations or noise. In our case, the data to consider consists of light curves composed of magnitude and phase, with a different number of points for each light curve. It involves finding a function that passes through the given data points while minimizing the overall curvature or roughness of the curve. This technique is particularly useful in situations where the data may contain random variations or noise, allowing for a clearer representation of underlying trends or patterns. The function for a *smoothing spline* typically involves minimizing the sum of squares of the deviations of the fitted curve from the data points, subject to a constraint on the overall curvature of the curve. Mathematically, the function can be represented as:(2)$$\sum_{i=1}^{n}{\left(y_{i}-f\left(x_{i}\right)\right)}^{2}+\lambda \int {\left(f^{\prime \prime }\left(x\right)\right)}^{2}\;dx$$
where f(x) is the smooth function being fitted to the data points; $x_{i},y_{i}$ are the data points; λ is a smoothing parameter that controls the trade-off between fidelity to the data and smoothness of the curve; and $\int {\left(f^{\prime \prime }\left(x\right)\right)}^{2}\;dx$ represents the integrated squared second derivative of the function, which penalizes high curvature. As shown in Figure 2, this is the set of light curves that represent the dataset pre-processed using the aforementioned method.

Figure 3 illustrates the distribution of metallicity within the dataset, highlighting a significant imbalance. Their metallicity distributions exhibit pronounced peaks around −1.5 dex, with relatively subdued tails on both the metal-rich and metal-poor ends. This is not due to a bias in our dataset, but it respects the metallicity distribution of RR Lyrae ab stars in our galaxy, as described in [7]. To address this imbalance, we introduced density-dependent sample weights for training our regression models. Specifically, we computed *Gaussian kernel density* estimates of the [Fe/H] distributions, assessed them for each object in the development sets, and assigned a density weight, $w_{d}$, to each data point, inversely proportional to the estimated normalized density.

Finally, for the predictive modeling of the [Fe/H] from the light curves, we use a dataset formed from the following two-dimensional sequences as input variables:(3)$$X^{<t>}=\left\{\begin{array}{c} m^{<t>}-<m>\\ Ph\cdot P \end{array}\right.t=\left\{1,...,N_{ep}\right\}$$
where $m^{<t>}$ is the magnitude of each data point, <m> is the mean magnitude, Ph is the phase, and *P* is the period. To verify the actual contribution of the preprocessing phase, three different datasets were created. The first dataset is based on phase-folded and phase-aligned data that have not undergone any preprocessing, which we will refer to for the sake of brevity as raw data. In this case, we have applied *padding* and *masking* methods to ensure the input tensor has the correct timestep shape. Padding is a specific form of masking where the masked steps are placed at the beginning or end of a sequence. It is necessary to pad or truncate sequences to standardize their length within a batch, allowing for the contiguous batch encoding of sequence data. Masking, on the other hand, informs sequence-processing layers that certain timesteps in the input are missing and should be ignored during data processing. The second dataset is derived from the raw data with the application of the smoothing spline method, but without incorporating the mean magnitude as described in Equation (Equation 3). The third dataset consists of the fully pre-processed data.

## 4. Metodology

This section explores model selection and optimization choices applied to deep-learning models to predict metallicity values from photometric light curves. Furthermore, a detailed description of each deep-learning model follows. In total, nine models were tested.

### 4.1. Model Selection and Optimization

Optimizing a predictive model involves two critical phases: *training* and *hyperparameter optimization*. During the *training phase*, the model’s parameters are fine-tuned by minimizing a cost function using a dedicated training dataset. For neural networks, this optimization typically employs gradient descent-based algorithms, leveraging the explicit expression of the cost function gradients. Hyperparameters, which dictate the model’s complexity—such as layer count, neuron density, and regularization methods—are predefined during the *training phase*. Their optimal configurations are determined by maximizing a performance metric on a separate validation dataset; ensuring the model’s effectiveness extends beyond the training data. Indeed, the *GridSearchCV* method from *Scikit-learn* was exploited for each of the nine created architectures. The hyperparameters explored included dropout rates of [0.1, 0.2, 0.4, 0.6], learning rates of [0.001, 0.01, 0.1], and batch sizes of [32, 64, 128, 256, 512]. For the *training phase*, we used the *mean squared error* (MSE) cost function with sample weights, as discussed in the previous section. To prevent overfitting the model to the training set, we experimented with different methods such as *kernel regularization (L1 and L2)* and *dropout layers*. By systematically evaluating the impact of various hyperparameters on model performance through the GridSearch tecnique, we aimed to identify the configuration that yields the highest accuracy and generalizability.

The metrics optimized during *hyperparameter tuning* must precisely mirror how effectively the trained model performs on unseen data, such as the validation set or test set. *Root mean squared error* (RMSE), *mean absolute error* (MAE), *weighted RMSE* (wRMSE), and *weighted MAE* (wMAE) were used as evaluation performance metrics. These metrics are essential tools for assessing the accuracy and performance of regression models, providing valuable insights into how well the models are performing and where improvements may be needed. Another objective is to minimize the mean prediction errors, a goal accomplished by maximizing the coefficient of determination ($R^{2}$ score).
(4)$$R^{2}=1-\frac{SS_{\mathrm{res}}}{SS_{\mathrm{tot}}}$$
where $SS_{\mathrm{res}}$ (residual sum of squares) is the sum of the squares of the residuals (the differences between the observed and predicted values). $SS_{\mathrm{tot}}$ (total sum of squares) is the total variability in the dependent variable, calculated as the sum of the squares of the differences between the observed values and the mean of the observed values. The $R^{2}$ value ranges from 0 to 1, with 1 indicating perfect prediction and 0 indicating that it does not explain any of the variability in the dependent variable. Higher $R^{2}$ values suggest a better model fit.

Moreover, cross-validation is essential for estimating the performance of a model on unseen data. We have chosen *repeated stratified K-fold cross-validation* that extends traditional K-fold cross-validation by incorporating both stratification and repetition. Through this validation method, the dataset is divided into K-folds while preserving the class distribution in each fold. This ensures that each fold is representative of the overall dataset, particularly crucial for imbalanced datasets, as in our case (see Figure 3). It further enhances robustness by repeating the process multiple times with different random splits. This helps to reduce the variance of the estimated performance metrics, providing more reliable insights into the model’s generalization capabilities. This method is particularly useful when dealing with small or imbalanced datasets, as in our specific case, where the performance estimation can be sensitive to the random partitioning of the data.

Moreover, we employed the *Adam* optimization algorithm with a *learning rate* of 0.01 for training each model, determining the optimal *early stopping* epoch based on the specific network type. To ensure a comprehensive representation of training examples across our entire [Fe/H] range, we utilized a sizable *mini-batch* equal to 256. All code is available within the open source GitHub repository (https://github.com/LorenzoMonti/metallicity_rrls, accessed on 4 July 2024).

### 4.2. Choosing the Right Neural Network Architecture for Time-Series Data

When working with time-series data, the choice of model architecture—convolutional neural networks (CNNs), recurrent neural networks (RNNs), or a mixed architecture combining both—depends on the specific characteristics of the data and the goals of the analysis. Each of these architectures has its strengths and weaknesses, and understanding these can help in choosing the most appropriate one for the required task.

CNNs are adept at detecting local patterns in data through convolutional filters. For time-series data, these patterns could be short-term trends or repeated cycles. Convolutional layers can automatically learn to identify important features such as peaks, troughs, and periodicity, which are useful for tasks like anomaly detection or classification. Unlike RNNs, CNNs do not rely on sequential processing, making them more efficient to train, especially on long sequences. This allows for the parallel processing of data, speeding up training and inference. Additionally, CNNs can capture hierarchical patterns by stacking multiple convolutional layers, which is beneficial for capturing complex patterns in time-series data that span different time scales. CNNs are commonly used in signal processing, anomaly detection, and time-series classification.

RNNs are designed to handle sequential data and maintain temporal dependencies through their recurrent connections, making them well-suited for tasks where the order of data points is crucial. Variants of RNNs like long short-term memory (LSTM) networks and gated recurrent units (GRUs) are capable of learning long-term dependencies, which are important for time-series data with long-term trends or patterns. RNNs can model the dynamics of time-series data over time, making them suitable for tasks like forecasting and sequential prediction. They are typically used in time-series forecasting, language modeling, and sequential prediction.

A mixed architecture leverages the strengths of both CNNs and RNNs: the feature extraction capabilities of CNNs and the temporal modeling abilities of RNNs. CNN layers can be used to extract local features from time-series data, which are then fed into RNN layers to capture temporal dependencies. Combining CNNs and RNNs can lead to improved performance on complex tasks by capturing both local patterns and long-term dependencies. This architecture is particularly useful when the time-series data has hierarchical patterns (e.g., short-term fluctuations and long-term trends). A mixed architecture provides flexibility in designing models tailored to specific tasks, such as multi-step forecasting, sequence classification, or anomaly detection. Example use cases for mixed architectures include multivariate time-series forecasting, complex sequence modeling, and hierarchical pattern recognition.

### 4.3. Network Architectures

In this section, the network architectures chosen and implemented—specifically convolutional neural networks (CNNs), recurrent neural networks, and their hybrid models—are explained, along with their functionality in the context of regression with time-series data. CNNs, while traditionally used for spatial data like images, can be adapted to capture local temporal patterns in time-series data, enhancing the feature extraction process. RNNs, designed to handle sequential data, excel at capturing temporal dependencies and trends crucial for accurate time-series forecasting. Finally, integrating CNNs and RNNs into hybrid models merges the strengths of both architectures.

#### 4.3.1. Fully Convolutional Network

A *fully convolutional network (FCN)* is designed for tasks like semantic segmentation [33] by using only locally connected layers, such as convolution, pooling, and upsampling layers, eliminating fully connected layers. This architecture reduces parameters, speeds up training, and handles varying input sizes. An *FCN*’s core structure includes a downsampling path for context capture and an upsampling path for spatial precision, with skip connections to retain fine spatial details. *FCNs* excel in pixel-level predictions, transforming classification networks into fully convolutional ones that produce dense output maps, and optimizing computations through overlapping receptive fields for efficient feedforward and backpropagation across entire images. *FCNs* are not limited to image data; they can also be effectively applied to time-series data for various regression and classification tasks. The architecture of *FCNs* allows them to capture temporal dependencies and patterns within time-series data by using convolutional layers to process sequences of data points. For time-series data, *FCNs* adapt the convolutional layers to operate along the temporal dimension, capturing local dependencies and patterns within the sequence. The core elements of an *FCN* for time-series analysis are as follows: (i) *Temporal convolutional layers*, which apply convolutional filters along the temporal axis to extract features from the sequence data. Using multiple filters enables the network to capture diverse aspects of the time series. (ii) *Pooling layers*, which reduce the temporal dimension by summarizing the information and lowering the computational burden.

In our architecture, the foundational unit consists of a *1D convolutional layer*, succeeded by a *batch normalization* layer [34] and an *ReLU activation* layer. The convolution is performed using three 1D kernels of sizes 8, 5, and 3, applied without any striding. Hence, the convolutional block is structured as *convolution layer* → *batch normalization layer* → *ReLU activation layer*. This arrangement allows the network to effectively extract and normalize features from the input time-series data while maintaining non-linearity. More formally, the foundational unit consists of:(5)$$\begin{array}{cc} y & =W⊗x+b\\ s & =\mathrm{BN}(y)\\ h & =\mathrm{ReLU}(s) \end{array}$$
where ⊗ is the convolutional operator. The implementation on the final network is based on stacking three convolutional blocks, each containing filters of sizes 128, 256, and 128, respectively. Following these convolutional blocks, the extracted features are passed through a *global average pooling* layer, significantly reducing the number of weights compared to a fully connected layer. The final output for regression is generated using a *dense* layer with a linear activation function.

#### 4.3.2. Inceptiontime

*InceptionTime* is a deep-learning architecture tailored for time-series regression, classification, and forecasting, inspired by Google’s Inception architecture [35]. It incorporates multiple parallel convolutional layers within its modules, akin to Inception, to capture features at diverse scales. These layers are adept at discerning both short-term and long-term patterns in the time-series data. Additionally, *InceptionTime* often integrates dilated convolutions to extend the receptive field without significantly increasing parameters. This feature proves valuable in capturing temporal dependencies over extended time spans. Residual connections are sometimes employed within the *InceptionTime* architecture. These connections facilitate gradient flow during training and mitigate the vanishing gradient problem, particularly in deeper networks. Furthermore, *temporal pooling* layers, such as *global average pooling*, are commonly used to aggregate information across the temporal dimension before making predictions. This step reduces data dimensionality while preserving essential temporal information. *InceptionTime’s* strength lies in its scalability and adaptability. It can be tailored to various time-series tasks and can accommodate time-series data of different lengths and sampling rates.

The architectural framework draws inspiration from the seminal works of Fawaz et al. (2019, 2020) [36,37], where they introduced the groundbreaking *InceptionTime* network. This network represented a substantial advancement over existing deep-learning models, achieving competitive performance comparable to the state-of-the-art TSC model. The architecture of *InceptionTime* revolves around the integration of two distinct residual blocks, strategically interconnecting the input and subsequent block inputs to address the challenge of vanishing gradients. Each residual block is structured with three Inception modules, each comprising two key components. The first component entails a bottleneck layer that serves a dual purpose: reducing the dimensionality of the time series using *m* filters and enabling *InceptionTime* to employ filters ten times longer than those in *Residual Network*, as elucidated in [37]. The second component involves the application of multiple filters of varying lengths to the output of the bottleneck layer. Simultaneously, a *max-pooling* layer is applied to the time series in parallel with these processes. The outputs from both the convolution and *max-pooling* layers are concatenated to form the output of the *Inception* module. Finally, *global average pooling* is applied to the final residual block, followed by propagation to a *dense* layer for regression analysis.

#### 4.3.3. Residual Network

*ResNet*, short for Residual Network, is a deep neural-network architecture that has revolutionized the field of computer vision since its introduction by Kaiming He et al. [38] in their paper “Deep Residual Learning for Image Recognition” in the *ImageNet Large-Scale Visual Recognition Challenge 2015* (ILSVRC2015). At its core, *ResNet* addresses the problem of vanishing gradients and degradation in training deep neural networks. As networks become deeper, they tend to suffer from diminishing performance gains and may even degrade in accuracy due to difficulties in optimizing the network’s parameters. This phenomenon arises because deeper networks are more prone to the vanishing gradient problem, where the gradients become increasingly small during backpropagation, hindering effective training. To overcome this challenge, *ResNet* introduces skip connections, also known as *residual connections*, that directly connect earlier layers to later layers. These skip connections allow the network to bypass certain layers, enabling the direct flow of information from the input to the output. By doing so, *ResNet* mitigates the vanishing gradient problem and facilitates the training of extremely deep networks. The key innovation of *ResNet* lies in its residual blocks, which consist of several convolutional layers followed by identity mappings or shortcut connections. These residual blocks enable the network to learn residual functions, capturing the difference between the desired output and the input to the block. This residual-learning approach enables the network to focus on learning the residual features, making it easier to optimize and train deep networks effectively. While ResNet is often associated with classification tasks, it can seamlessly adapt to regression tasks by utilizing appropriate loss functions such as *mean squared error* (MSE) or *mean absolute error* (MAE). These loss functions quantify the discrepancy between predicted and actual continuous values, guiding the training process toward minimizing prediction errors.

In our architectural design, we repurpose the foundational units established in Equation (Equation 5) to assemble every residual block. Let $Block_{k}$ denote the foundational unit with *k* filters. The formulation of the residual block is thus articulated as follows:(6)$$\begin{array}{cc} h_{0} & ={\mathrm{Block}}_{k_{0}}\left(x\right)\\ h_{1} & ={\mathrm{Block}}_{k_{1}}\left(h_{0}\right)\\ h_{2} & ={\mathrm{Block}}_{k_{2}}\left(h_{1}\right)\\ y & =x+h_{2}\\ \hat{h} & =\mathrm{ReLU}(y) \end{array}$$
where the number of filters $k_{i}$ = {64, 64, 64}. The ultimate Residual Network configuration comprises three sequential residual blocks, succeeded by a *global average pooling* layer and a *dense layer* with a linear activation.

#### 4.3.4. Long Short-Term Memory and Bi-Directional Long Short-Term Memory

The *LSTM-based* models are an extension of RNNs, which are able to address the vanishing gradient problem in a very clean way. The *LSTM* models essentially extend the RNNs’ memory to enable them to keep and learn long-term dependencies of inputs [39,40]. This memory extension has the ability to remember information over a longer period of time and thus enables reading, writing, and deleting information from their memories. The *LSTM* memory is referred to as a “gated” cell, a term inspired by its capability to selectively retain or discard memory information. An *LSTM* model captures important features from inputs and preserves this information over a long period of time. The decision of deleting or preserving the information is made based on the weight values assigned to the information during the training process. Hence, an *LSTM* model learns what information is worth preserving or removing. A variant of the *LSTM* architecture is the *bidirectional LSTM* (*BiLSTM*) network, which incorporates information from both past and future time steps. *BiLSTM* consists of two *LSTM* layers, one processing the input sequence in the forward direction and the other processing it in the backward direction. This bidirectional processing allows the network to capture context from both preceding and subsequent time steps, enhancing its ability to model complex dependencies in sequential data.

In our architectures, the main block consists of an *LSTM* or bidirectional *LSTM* layer, succeeded by a *dropout layer* to prevent overfitting and improve the network’s ability to generalize, making it a crucial technique for enhancing the performance and robustness of models dealing with sequential data. The ultimate *LSTM* or Bi-LSTM network configuration includes three main blocks, succeeded by a *dense* layer with a linear activation, as formally described below:(7)$$\begin{array}{cc} h_{t}^{\left(0\right)} & =X\\ for\;i & =1\;\mathrm{to}\;n\\ \; & h_{t}^{\left(i\right)}=RNN\_block\left(d_{t}^{\left(i-1\right)}\right)\\ \; & d_{t}^{\left(i\right)}=Dropout\left(h_{t}^{\left(i\right)}\right)\\ end & \;for\\ \hat{y}= & \mathrm{Dense}(d_{t}^{\left(n\right)}) \end{array}$$
where *n* is the number of the main blocks, which in our case is 3, and RNN_block depends on the type of model used (*LSTM* or *BiLSTM*). To avert overfitting the model to the training set, we tried different techniques, including the *kernel regularization* method and *recurrent regularization*. In the *LSTM* architecture, *kernel regularization* parameters are set to l1 = 0 and l2 = $2\cdot 10^{-6}$, while the *recurrent regularization* parameters are set to l1 = $2\cdot 10^{-6}$ and l2 = 0. In the *BiLSTM* version, both kernel and *recurrent regularization* parameters are set to l1 = $2\cdot 10^{-6}$ and l2 = 0. Each *dropout* layer has a rate of 0.2, 0.2, and 0.1, respectively. Since the input tensor in RNNs must always have the same shape and the raw light curves have a varying number of points, the technique known as *padding-masking* was used to train the unprocessed dataset described in Section 3. In practical terms, this is a crucial technique for handling variable-length sequences in neural networks, particularly those dealing with sequential data like time series or text. It ensures that padded values (−1 at the end of the time series, in our case), which are added to sequences to make them uniform in length, do not affect the model’s learning process.

#### 4.3.5. Gated Recurrent Unit and Bi-Directional Gated Recurrent Unit

The *GRU* (gated recurrent unit) network, introduced by Kyunghyun Cho et al. [41], is a type of recurrent neural-network (RNN) architecture that excels in modeling sequential data like time-series or natural language sequences. Unlike traditional RNNs, but similar to the *LSTM* network described above, the *GRU* incorporates gating mechanisms to regulate information flow within the network, allowing it to selectively update its internal state at each time step. This architecture comprises recurrent units organized in layers, with each unit processing input sequences step by step while maintaining an internal state representation capturing information from previous steps. One of the key advantages of *GRU* lies in its gating mechanisms, which include the update gate and the reset gate.

The update gate determines the amount of previous state information to be maintained and the amount of new information to be incorporated from the current input, while the reset gate determines the amount of past information to be forgotten when computing the current state. These gates enable the network to adaptively update its internal state, making it adept at capturing long-term dependencies in sequential data. A variant of the *GRU* architecture is the bidirectional GRU (*BiGRU*), which incorporates information from both past and future time steps. *BiGRU* consists of two *GRU* layers, one processing the input sequence in the forward direction and the other processing it in the backward direction. This bidirectional processing allows the network to capture context from both preceding and subsequent time steps, enhancing its ability to model complex dependencies in sequential data.

The architectures, in this case, are completely similar to those described in *LSTM* and *BiLSTM*. The substantial change concerns the main blocks, in fact, they are made up of *GRU* or *BiGRU* layers, respectively.

#### 4.3.6. Convolutional GRU and Convolutional LSTM

*Convolutional LSTM* and *convolutional GRU* architectures integrate convolutional layers with *LSTM* (long short-term memory) or *GRU* (gated recurrent unit) layers to leverage the strengths of both convolutional and recurrent networks. These mixed architectures are particularly useful for tasks involving sequential data with spatial dimensions, such as video processing, time-series analysis, and spatio-temporal forecasting [42,43,44]. In these architectures, convolutional layers are typically used to extract spatial features from input data. These layers apply filters across the input to detect local features like edges, textures, and shapes, generating feature maps that summarize spatial information. After the convolutional layers, *LSTM* or *GRU* layers are introduced to capture the temporal dependencies in the data. In a typical convolutional *LSTM* or *GRU* architecture, the input data are first processed by a series of convolutional layers, which may include *pooling* layers to reduce the spatial dimensions and capture hierarchical features. The output of these convolutional layers is then reshaped to match the input requirements of the *LSTM* or *GRU* layers, which process the data sequentially to model temporal relationships. This hybrid approach is particularly effective for time-series data, which often contains intricate temporal dependencies alongside potentially complex patterns within individual time steps.

Our architecture is divided into two primary blocks: (i) a *convolutional block* and (ii) a *recurrent block*. The convolutional block is based on the fully convolutional network (*FCN*) architecture discussed in Section 4.3.1, particularly in Equation (Equation 5). It includes a *1D convolutional layer* followed by a *batch normalization* layer and a *ReLU activation* layer. The convolution uses three 1D kernels of sizes 8, 5, and 3, applied without any striding. The final configuration of this block involves stacking three convolutional layers with filter sizes of 128, 256, and 128, respectively. After these convolutional layers, a *1D max pooling* layer is applied to reduce the spatial dimensions and capture hierarchical features. The recurrent block includes an *LSTM* or *GRU* layer, depending on the specific architecture, followed by a *dropout* layer to prevent overfitting and enhance the network’s generalization ability. This block stacks two such layers. To further mitigate overfitting, we exploited various techniques, including *kernel regularization* and *recurrent regularization*. The *kernel regularization* parameters are set to l1 = 0 and l2 = $2\cdot 10^{-6}$, while the *recurrent regularization* parameters are set to l1 = $2\cdot 10^{-6}$ and l2 = 0. Additionally, the *dropout layers* have rates of 0.2 and 0.1, respectively. The final architecture includes a *dense* layer with a linear activation function.

## 5. Results and Discussion

### 5.1. Experiment Setup

The foundational system utilized in our training is a workstation equipped with an NVIDIA GeForce RTX 4070 GPU. For software implementations, we employ Python 3.10, TensorFlow 2.13.0, Keras 2.13.1, and CuDNN 11.5 libraries.

### 5.2. Results of the Experiments

The regression performance of our finalized pre-processed models underwent evaluation using a range of standard metrics applied to the stratified K-fold validation datasets. Notably, the $R^{2}$ metric was treated uniquely, with the mean value being computed. These metrics collectively suggest a minimal generalization error, indicative of strong predictive capabilities on unseen data. Within Table A1, these metric values are showcased for the G band, juxtaposed with corresponding metrics from the training data, serving as a reference point. The striking resemblance between the training and validation dataset metrics underscores a commendable balance between bias and variance in our completed pre-processed models. Moreover, the standard deviation was calculated (and presented in Table A1) for each value to determine the confidence intervals.

Furthermore, the efficacy of the preprocessing step in enhancing model performance is discernible. As illustrated in Table A1, the (i) *raw catalog* exposes instances of overfitting in CNN models, likely attributable to an excessively detailed fit to the training data, which fails to generalize well to novel instances. Conversely, the RNN models, particularly those based on GRU architecture, exhibit superior performance, with *LSTM-based* models demonstrating divergence. Indeed, while both *LSTM* and GRU networks benefit from data preprocessing, *LSTMs* are generally more sensitive to the quality of the input data and hence gain more from the preprocessing steps. GRUs, with their simpler architecture, can sometimes handle less-pre-processed data more robustly but still show improvements with proper preprocessing. Employing the (ii) *pre-processed catalog without mean magnitude*, yields improved and more stable performance across datasets. Nonetheless, it is the (iii) *pre-processed catalog* that yields optimal outcomes, with GRU emerging as the best model among them. This reaffirms that the preprocessing step has significantly upgraded the models in terms of performance, demonstrating its essential role in improving the predictive accuracy and robustness of deep-learning models.

Figure 4 shows the schematic architecture of our best-performing predictive model, while its main parameters and hyperparameters are listed in Table 2. More in-depth, as already mentioned in Section 4.3.5, it contains three blocks based on *GRU* layers, utilizing L1 regularization on the *recurrent regularizer* parameters and L2 regularization on the *kernel regularizer* parameters. Additionally, *dropout* layers are employed after every *GRU* layer to prevent overfitting and enhance generalization. This architecture was specifically designed to balance complexity and performance, ensuring robust predictions while maintaining computational efficiency. The strategic placement of regularization and *dropout* layers effectively mitigates the risk of overfitting, contributing to the superior performance of our models on unseen data. The combination of these architectural choices and hyperparameter configurations enables our *GRU-based* model to capture temporal dependencies effectively, handle varying sequence lengths, and maintain stability during the training step.

The training step of our model involved a comprehensive cross-validation approach to ensure robust and reliable performance. We employed a 5 K-fold cross-validation strategy to evaluate the generalization capabilities of our model. During each fold, the dataset was split into training and validation subsets, with the model being trained on the training subset and evaluated on the validation subset. The primary goal of the training step was to minimize the MSE loss, a metric that captures the performance of the model by penalizing incorrect predictions.

As shown in Figure 5, the training loss (red) and validation loss (green) are plotted across the epochs for each of the five cross-validation folds. The plot demonstrates a steady decrease in both training and validation loss over the training epochs, indicating the model’s learning process. The consistency between training loss and validation loss across the folds (the darker the color, the more similar the results were between the folds) suggests that the model is not overfitting to the training data. Instead, it generalizes well to unseen data, as evidenced by the closely aligned loss values. This alignment between training and validation performance underscores the effectiveness of our model architecture and the preprocessing steps implemented to enhance data quality and model robustness. Overall, the training step was successful in developing a predictive model that maintains high performance on validation data. As presented on the left side of Figure 6, the histograms of the ground-truth metallicity values in red, and predicted in gray [Fe/H] values from our best G-band model for the training (on top) and validation (on bottom) datasets. The histogram for the training data exhibits a close alignment between the ground-truth and predicted [Fe/H] values, indicating high predictive accuracy and successful learning of the metallicity distribution. Similarly, the validation dataset histogram shows good overlap, with only slight variability, suggesting robust generalization to unseen data. Moreover, the right side of Figure 6 shows the predicted versus the true photometric metallicities from the *GRU* predictive model. The top and bottom panels display the full training and validation datasets, respectively, with the red lines denoting the identity function. The scatter plot for the training data (displayed on the top-right panel of Figure 6) indicates that the predicted [Fe/H] values closely follow the identity line (y=x), confirming the model’s high accuracy. The validation data scatter plot (displayed in the bottom-right panel of Figure 6) shows a similarly strong correlation with minor dispersion, affirming the model’s capability to generalize well to new data. These analyses highlight the *GRU-based* model’s robust performance in predicting photometric metallicities. The histograms and scatter plots demonstrate that the model maintains high accuracy and generalizes effectively beyond the training dataset.

### 5.3. Performance Comparison of Predictive Models

The prediction of photometric metallicity of RR Lyrae stars is a crucial task in astrophysics, offering insights into the chemical composition and evolution of stellar populations. RR Lyrae stars are particularly valuable as they are abundant in old stellar populations and have well-defined relationships between their light curve characteristics and metallicity. Traditional methods for estimating metallicity often rely on empirical formulas derived from the Fourier parameters of the light curves, which can introduce biases due to the heterogeneity and noise in the observational data. Despite the importance of this task, the application of state-of-the-art algorithms in predicting photometric metallicity based exclusively on light curves remains relatively scarce. Traditional methods, predominantly based on empirical relations and classical statistical approaches, have been widely used but often fall short in capturing the complex, non-linear relationships inherent in stellar light curves. In recent years, state-of-the-art algorithms, particularly deep-learning models, have shown significant promise in addressing these challenges. In this regard, the paper used for the performance comparison is that of Dekány (2022) [29]. The metrics are taken directly from the aforementioned paper and are those related to the light curves of RR Lyrae stars in Gaia G band, as used in our work. Table 3 summarizes the key differences and similarities between the two models, providing a clear and concise comparison of their performance and design. Upon analyzing the performance metrics, it is evident that our GRU model outperforms the BiLSTM model by Dekány et al. in nearly all metrics. Based on the metrics provided in Table 3, the proposed best model, which uses a GRU architecture, outperforms the Dekány best model, which employs a BiLSTM architecture, in almost all aspects. When examining the *R*^2^ regression performance, the BiLSTM model shows a slightly better performance on the training set, with an *R*^2^ of 0.96 compared to the GRU model’s 0.9447. In terms of weighted root mean squared error (wRMSE), the GRU model demonstrates lower values for both the training and validation sets (0.0733 and 0.0763, respectively) compared to the BiLSTM model (0.10 and 0.13). This indicates that the GRU model has more accurate predictions. Similarly, for weighted mean absolute error (wMAE), the GRU model shows better performance, with training and validation values of 0.0547 and 0.0563, respectively, while the BiLSTM model has values of 0.07 and 0.10. Looking at the root mean squared error (RMSE), the GRU model again exhibits significantly lower values for both training (0.0735) and validation (0.0765) compared to the BiLSTM model (0.15 and 0.18). This reinforces the superior overall performance of the GRU model. The mean absolute error (MAE) metrics also support this conclusion, with the GRU model showing lower values for training (0.0549) and validation (0.0565) compared to the BiLSTM model’s 0.12 and 0.13.

## 6. Conclusions

In this study, we demonstrated the application of deep-learning models to predict the photometric metallicity of RR Lyrae stars using G-band time series published in Gaia DR3. By leveraging advanced neural-network architectures, we successfully developed predictive models that can accurately estimate metallicity, achieving a low *mean absolute error* of 0.056 dex and a high $R^{2}$ regression performance of 0.9401 measured by cross-validation. The *GRU-based* model, among the various deep-learning algorithms tested, exhibited superior performance in predicting photometric metallicities, underscoring the potential of deep learning in handling complex relationships within astronomical datasets. The histograms of true and predicted [Fe/H] values for both training and validation sets indicate a good alignment between predicted and actual values. The scatter plots of predicted versus true photometric metallicities further validate the model’s predictive accuracy. The proximity of data points to the identity line (red line in right panels of Figure 6) in both training and validation sets demonstrates the model’s robustness. Despite these advancements, our work also emphasizes the ongoing challenges in deep-learning applications for astronomical data, such as handling data heterogeneity and ensuring model generalization across different datasets. Future work aims to expand the scope of our study by incorporating different datasets, such as those involving first overtone RR Lyrae stars (type c; RRc) to further test and validate our models. Additionally, we plan to investigate the potential of other advanced models, such as Transformer architectures, which have shown great promise in handling sequential data and could potentially offer improvements over the current models. By exploring these new datasets and models, we hope to enhance the predictive capabilities and generalization performance of our deep-learning approaches, thereby providing more accurate and comprehensive insights into the metallicities of RR Lyrae stars.

In conclusion, the integration of deep-learning techniques into the field of astrophysics presents a promising avenue for enhancing our understanding of stellar populations and various aspects of variable stars. The methodologies and findings from this study contribute to the broader efforts of leveraging big data in astronomy, paving the way for more precise and comprehensive analyses of astrophysical phenomena.

## Figures and Tables

**Figure 1 sensors-24-05203-f001:**
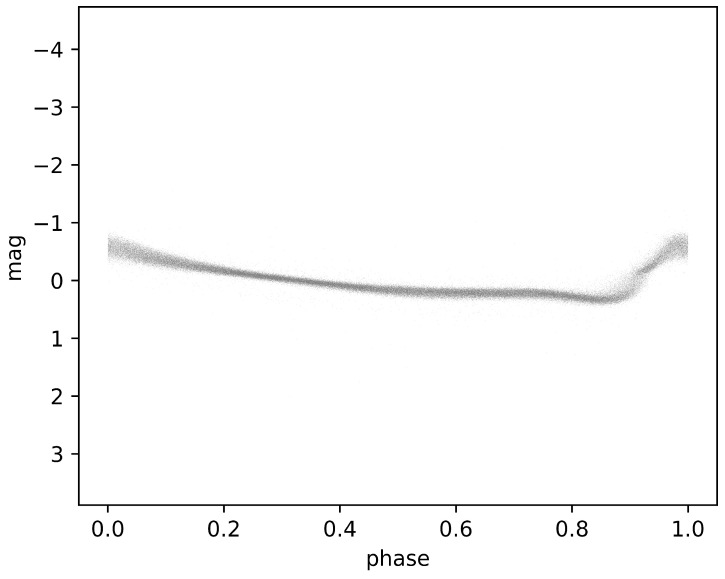
Phase-folded and phase-aligned G-band light curves of 6002 RRab Stars. The two-dimensional plot depicts the phase and magnitude of G-band light curves for all the RRab stars, showcasing their characteristics after phase-folding and alignment.

**Figure 2 sensors-24-05203-f002:**
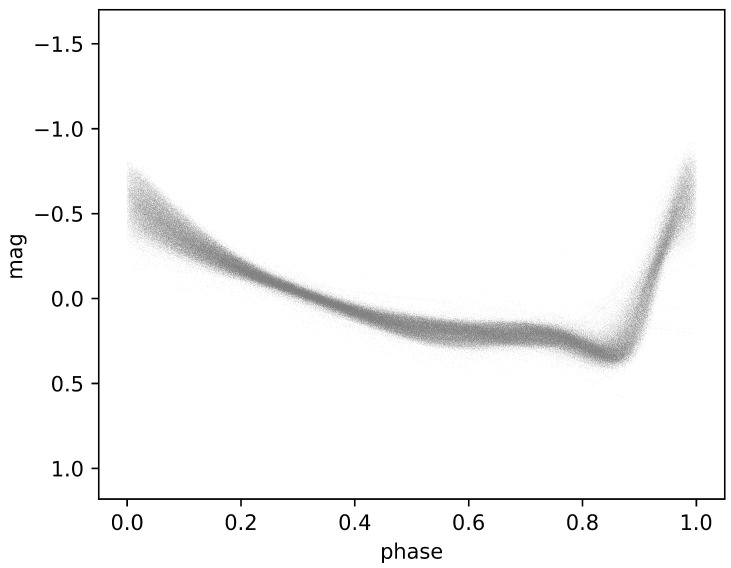
The two-dimensional representation illustrates the *phase* and *magnitude* of *G-band* light curves following the application of the smoothing spline method.

**Figure 3 sensors-24-05203-f003:**
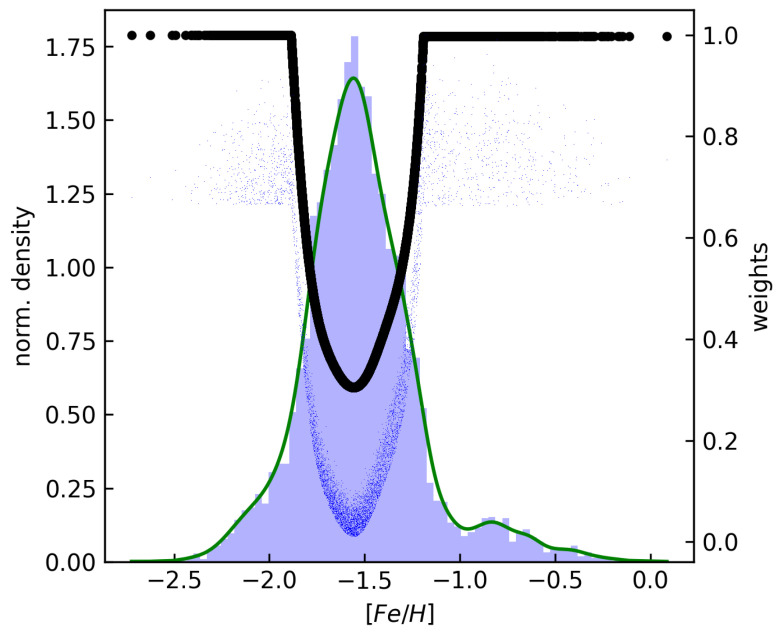
The distribution of metallicity of 6002 RRab stars in our dataset along with their respective sample weights is illustrated. Histograms are marked by blue bars, while kernel density estimates of the [Fe/H] values are represented by green curves. Black symbols denote the (normalized) weights derived from the inverse of the density. The final sample weights are denoted by blue points.

**Figure 4 sensors-24-05203-f004:**
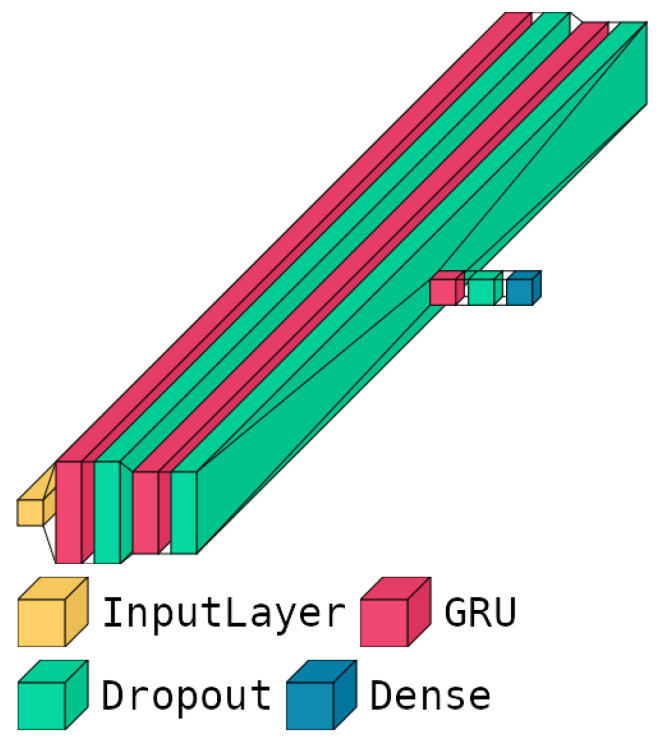
Graphic representation of the GRU model. The picture illustrates the layered structure of the GRU model, detailing the arrangement and interactions of the GRU layers, including input, hidden, and output layer (dense layer).

**Figure 5 sensors-24-05203-f005:**
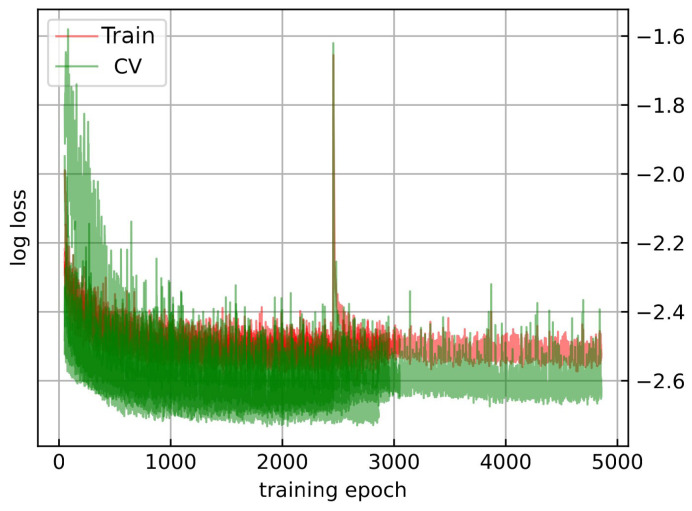
Training loss (red) versus validation loss (green) for each cross-validation fold (5). The plot illustrates the consistency between training and validation performance, indicating the model’s ability to generalize well.

**Figure 6 sensors-24-05203-f006:**
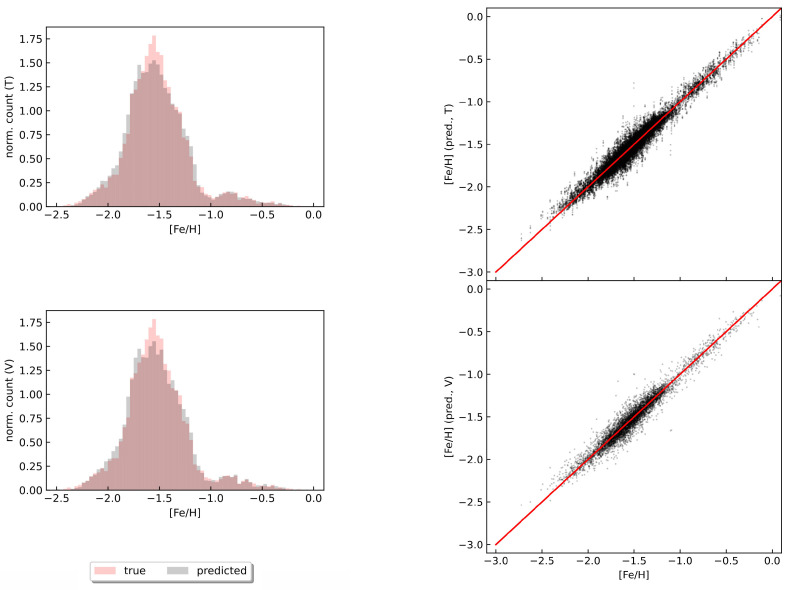
On the **left** side are presented histograms of the true (red) and predicted (gray) [Fe/H] values from our best G-band model for the training (T; **top**) and validation (V; **bottom**) datasets. On the **right** side are shown true vs. predicted photometric metallicities from the *GRU* predictive model. The **top** and **bottom** panels show the full training (4801 time-series) and validation datasets (1201 time-series), respectively. The red lines denote the identity function.

**Table 1 sensors-24-05203-t001:** Parameters of 6002 RRab stars from the Gaia DR3 catalogue (Clementini et al., 2023) [15]: (1) identification number; (2) Gaia DR3 source_id; (3) Pulsation period (days); (4) Amplitude in the G band; (5) Number of epochs; (6)–(7) Photometric metallicity and errors from Muraveva et al. (2024) [32] ( photometric metallicity values have a range from −3 to 1).

id	Source_id	Period	AmpG	#Epochs	[Fe/H]	σ[Fe/H]
0	5978423987417346304	0.415071	0.61029154	53	−0.144963	0.398111
1	5358310424375618304	0.407642	0.6174223	56	−0.223005	0.391468
2	5341271082206872704	0.327778	0.7399841	53	0.087612	0.382031
3	5844089608021904768	0.459576	0.47177884	54	−0.380516	0.396500
4	5992931321712867200	0.390948	0.76943225	63	−0.256892	0.391830
...	...	...	...	...	...	...
5997	5917421845281955584	0.532958	0.9153245	52	−1.507490	0.379151
5998	4659766188753815552	0.413777	0.9984105	245	−0.758832	0.373709
5999	5868263951719014528	0.365109	1.0959375	66	−0.300124	0.380579
6000	5963340573264428928	0.452752	1.0733474	64	−1.079237	0.369655
6001	5796804423258834560	0.510323	1.0356201	57	−1.553741	0.372771

**Table 2 sensors-24-05203-t002:** Parameters and hyperparameters of the best G-band predictive model (GRU). The table details the configuration of the GRU deep-learning model, including the number of layers, specific hyperparameters, and the model parameters.

Layers	Hyperparameters	Parameters
input_1	...	0
gru_1	20 units	1440
dropout_1	$P_{d}$ = 0.2	0
gru_2	16 units	1824
dropout_2	$P_{d}$ = 0.2	0
gru_3	8 units	624
dropout_3	$P_{d}$ = 0.1	0
dense	...	9

**Table 3 sensors-24-05203-t003:** Comparison of performance metrics for predictive models: proposed best model vs. Dekány (2022) best model. The table includes common metrics from both studies: *R*^2^, RMSE, MAE, wRMSE, and wMAE. These metrics were computed for both the training and validation phases.

Metrics		Proposed Best Model (GRU)	Dekány Best Model (BiLSTM)
Dataset (number of RR Lyrae stars)		6002	4458
*R*^2^ regression performance	training	0.9447	0.96
	validation	0.9401	0.93
wRMSE	training	0.0733	0.10
	validation	0.0763	0.13
wMAE	training	0.0547	0.07
	validation	0.0563	0.10
RMSE	training	0.0735	0.15
	validation	0.0765	0.18
MAE	training	0.0549	0.12
	validation	0.0565	0.13
Architecture		GRU (3 layers)	BiLSTM (2 layers)
Regularization		L1, L2	L1
Dropout		Yes	Yes

## Data Availability

All code is open-source and available within the https://github.com/LorenzoMonti/metallicity_rrls (accessed on 4 July 2024) GitHub repository. The dataset can be downloaded on the Gaia Archive site, on the Gaia DR3 page according to the selection criteria presented in Section 3.

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
