# Peer review of "Leveraging Deep Learning for Time-Series Extrinsic Regression in Predicting the Photometric Metallicity of Fundamental-Mode RR Lyrae Stars"

_sensors, 2024, doi:10.3390/s24165203_

Round 1

Reviewer 1 Report

Comments and Suggestions for Authors

Referee report on “Leveraging Deep Learning for Time Series Extrinsic Regression in predicting photometric metallicity of Fundamental-mode RR Lyrae Stars”

by Lorenzo Monti, Tatiana Muraveva, Gisella Clementini, Alessia Garofalo

   The referred paper is interesting and presents the author’s new results. The authors apply deep learning techniques, particularly advanced neural network architectures, to study the metallicity and probably to predict a photometric metallicity of RR Lyrae stars. They are using observational data from ESA’s Gaia telescope, published in Gaia DR3.

In this paper, the authors present their developed models by which to estimate metallicity in RR Lyraes, achieving a low mean absolute error of 0.056.

The authors emphasize on the potential and advantages of deep learning applications for astronomical data and photometric surveys.

Some general comments:

1. Authors made an extensive review of deep learning, which they included in the Section 2. Background. This could help the readers to understand the applied methods and techniques. The text is written well with very detailed explanations. Although, there are very long sentences in some places of the text such that the main sense could be lost. I recommend some of them to be divided into two/three sentences to be clearer and more understandable.

2. I found some of the expressions sound not very scientifically.  This should be checked carefully by the authors and replace them with suitable words.

Methods and models:

 ·      The methods are described in details. Are the applied methods concerned to all RR Lyras or are they only suitable to some of the stars of this type? Maybe this should be mentioned.

Figures:

 Figures 1,2,3 – the dpi should be increased and if it is possible to saturate the colors of the light curves. They need to be better visible.

References:

 ·       Numbers [10] and [27] in the Reference’s list do not have full descriptions. Please add this.

Minor comments:

Comment 1. It is not very clearly shown what RRab stars present. Insert the definition or shortly (1-2 sentences) describe them.

Comment 2. Row 22: “such as distance” – Is that a distance to the concrete objects or to all RR Lyrae stars?

Comment 3. Rows 108 – 115:  Is all this text concerns to the paper citation [21]? It is a little bit misunderstanding which is the method and about the methodology, and are they studied in the same paper?:  “It generates a prioritized catalog of both overarching and localized anomalies. The method computes anomaly scores for each light curve concerning a cluster of centroids derived from a modified k-means clustering algorithm. The methodology achieves scalability for large datasets by employing sampling techniques. The performance has been validated on various datasets, including light-curve data, showcasing its proficiency in identifying known anomalies.”

Comment 4. Table 1: Pulsation period: Give the units (days, hours, etc…) in the table’s title or in the table column;

Comment 5. Table 1: id 2, [Fe/H] is 0.087612 – is it something different in this sample? It is look like the only positive value? Explain, especially if it has some unordinary result.

Comment 6. Row 201: ls it really that the “final sample consists of 6002 RRab stars”?

Comment 7. Equation 1 – It is not clear, if this formula is obtained by the authors or it is from the literature. Please, specify!

Comment 8. Row 280: “each deep learning model (9)” – it is not very clear what (9) shows? Please clarify!

Comment 9. Table A1 – There are different decimal signs used in the values: commas “,” and dot “.”. Is it for a reason? Or just a technical mistake?

Comments on the Quality of English Language

The text need some minor corrections.

Author Response

Dear Reviewer,

Thank you for your thorough and insightful feedback on our manuscript. We appreciate the time and effort you have put into providing constructive comments. We have addressed each of your comments below and made corresponding revisions in the manuscript.

  1. Reviewer Comment 1: Authors made an extensive review of deep learning, which they included in the Section 2. Background. This could help the readers to understand the applied methods and techniques. The text is written well with very detailed explanations. Although, there are very long sentences in some places of the text such that the main sense could be lost. I recommend some of them to be divided into two/three sentences to be clearer and more understandable.

  • In response to your comment about the length of some sentences, we have made revisions to improve clarity. Specifically, we have divided long sentences into shorter, more manageable ones to enhance readability and ensure that the main points are conveyed effectively. We have made the following changes:

    • Two sentences have been revised in the "Time-series Classification" section.

    • One sentence has been revised in the "Time-series Regression" section.

           Believing these changes address your concerns and improve the overall readability of the text.

  1. Reviewer Comment 2: I found some of the expressions sound not very scientifically.  This should be checked carefully by the authors and replace them with suitable words

  • In response to your comment, we have reviewed and revised the expressions in the manuscript to ensure that they meet the appropriate scientific standards. Specifically, we have made modifications in the following sections:

    • Section 5.2

    • Methodology, Section 4

  1. Reviewer Comment 3: The methods are described in details. Are the applied methods concerned to all RR Lyras or are they only suitable to some of the stars of this type? Maybe this should be mentioned.

  • As stated in the title of the paper, the models we discuss are specifically designed for fundamental-mode RR Lyrae stars (ab type). However, the methodologies we employed are generally applicable to any photometric light curves. This means that, in principle, the models could be extended to other types of variable stars, provided that they are retrained with appropriate hyperparameter tuning.

  1. Reviewer Comment 4: Figures 1,2,3 – the dpi should be increased and if it is possible to saturate the colors of the light curves. They need to be better visible.

  • We have addressed your comments regarding the resolution and visibility of the figures. Specifically:

    • Figures 1, 2, and 3 have all been updated to 1200 dpi.

    • The colors of the light curves have been enhanced to improve visibility, with the gray curves being saturated.

These changes should make the figures clearer and more visually effective. We appreciate your suggestion and believe these improvements address your concerns.

  1. Reviewer Comment 5: Numbers [10] and [27] in the Reference’s list do not have full descriptions.

  • We have updated the references [10] and [27] in the Reference list to include full descriptions. The revised entries now provide complete information as required.

  1. Reviewer Comment 6: It is not very clearly shown what RRab stars present. Insert the definition or shortly (1-2 sentences) describe them.

  • To address this, we have added a couple of sentences to the Introduction section of the manuscript. These sentences provide a brief definition and description of RR Lyrae stars, including RRab stars, to ensure that their characteristics are clearly conveyed.

  1. Reviewer Comment 7: “such as distance” – Is that a distance to the concrete objects or to all RR Lyrae stars?

  • To clarify, when we refer to "distance," we are specifically referring to the distance to each individual RR Lyrae star. We have revised the manuscript to make this distinction explicit.

  1. Reviewer Comment 8: Is all this text concerns to the paper citation [21]? It is a little bit misunderstanding which is the method and about the methodology, and are they studied in the same paper?:  “It generates a prioritized catalog of both overarching and localized anomalies. The method computes anomaly scores for each light curve concerning a cluster of centroids derived from a modified k-means clustering algorithm. The methodology achieves scalability for large datasets by employing sampling techniques. The performance has been validated on various datasets, including light-curve data, showcasing its proficiency in identifying known anomalies.”

  • To clarify, the text you referred to describes the PCAD method, which is specifically focused on anomaly detection in unsynchronized periodic time series. I have revised the relevant section to more accurately reflect the scope of the cited work. The paper discussed pertains to how the PCAD method generates a prioritized catalog of anomalies, employs a modified k-means clustering algorithm, and achieves scalability through sampling techniques. The updated section now clearly distinguishes between the methods and their applications, and explicitly states that the PCAD method, as presented in the cited paper, supports scalability for large datasets.

  1. Reviewer Comment 9: Table 1: Pulsation period: Give the units (days, hours, etc…) in the table’s title or in the table column;

  • We have added the unit of measurement to the table to clarify that the pulsation period is expressed in days. This information is now included in the table’s title for better clarity.

  1. Reviewer Comment 10: Table 1: id 2, [Fe/H] is 0.087612 – is it something different in this sample? It is look like the only positive value? Explain, especially if it has some unordinary result.

  • The value of 0.087612 is indeed within the expected range for photometric metallicities measured in the Gaia G band, which spans from approximately -3 to 1. This value is not unusual or erroneous; it simply represents a positive metallicity within the range of our sample. We have added a clarifying sentence to the table caption to explain that photometric metallicities are typically in the range from -3 to 1 and that the observed value is within this normal range.

  1. Reviewer Comment 11: Row 201: ls it really that the “final sample consists of 6002 RRab stars”?

  • Yes, the dataset used for training the models consists of 6002 RRab stars. While this number might seem modest, the results demonstrate that this sample size is sufficient for effectively training and validating our models. The performance metrics obtained validate the adequacy of the dataset for the intended analyses.

  1. Reviewer Comment 12: Equation 1 – It is not clear, if this formula is obtained by the authors or it is from the literature. Please, specify!

  • The formula presented in Equation 1 is a standard definition of phase commonly used in the literature. To clarify this, we have updated the text above the equation to specify that it is a widely recognized standard definition.

  1. Reviewer Comment 13: Row 280: “each deep learning model (9)” – it is not very clear what (9) shows? Please clarify!

  • We have added a clarifying sentence to the text and spelled out the reference number to ensure that it is clear what "(9)" refers to. This addition should help in understanding the context and source of the information.

  1. Reviewer Comment 14: Table A1 – There are different decimal signs used in the values: commas “,” and dot “.”. Is it for a reason?

  • The use of different decimal signs was indeed a technical mistake. We have corrected this inconsistency throughout the manuscript to ensure uniformity in decimal notation. Thank you for your careful review and for allowing us to rectify this issue.

We believe that these revisions have strengthened our manuscript, and we look forward to your further feedback. Thank you again for your valuable comments.

Sincerely,

Lorenzo Monti, PhD

National Institute for Astrophysics – OAS (Bologna)

Reviewer 2 Report

Comments and Suggestions for Authors

Summary

The study investigates the utilization of deep learning to estimate the metallicity of RR Lyrae stars by analyzing data from the Gaia mission. The research team devised a novel method for predicting metallicity from light curves in the Gaia optical G-band by utilizing deep learning models, specifically GRU-based architectures. The study's high R² regression performance of 0.9401 and low mean absolute error of 0.056 illustrate the efficacy of deep learning in the analysis of large astronomical datasets and the provision of precise insights into the metallicity of these stars.

Major Comments

The manuscript includes references to phase folding, alignment, and the application of smoothing splines, provide comprehensive details regarding managing outliers and missing values.

The manuscript does not provide a clear rationale for the selection of certain models over others.

The manuscript briefly discusses hyperparameter tuning but it does not provide any specifics regarding the search space, optimization techniques or the computational resources that were employed.

The manuscript does not include confidence intervals or statistical significance testing for metrics such as MAE and R². Kindly include.

The interpretability of the deep learning models is not addressed in the manuscript.

Minor Comments

The abstract could be enhanced by including more detailed information regarding the results, such as the precise performance metrics.

Publication years of certain important references in the introduction section are absent. 

Insufficient captions in certain figures and tables.

The manuscript contains minor grammatical errors.

The manuscript mentions the availability of code, but it does not provide a direct link.

Comments on the Quality of English Language

N/A

Author Response

Dear Reviewer,

We sincerely appreciate your detailed and thoughtful feedback on our manuscript. Your constructive comments have been invaluable, and we are grateful for the time and effort you dedicated to reviewing our work. We have carefully considered each of your suggestions and have made the necessary revisions to the manuscript, as detailed below.

  1. Reviewer Comment 1: The manuscript includes references to phase folding, alignment, and the application of smoothing splines, provide comprehensive details regarding managing outliers and missing values.

  • We have added the following details to address your concerns:

      • Missing Values: We have included additional information in the manuscript specifying that a check is performed to ensure that phase values are less than 1.0 and that period values are not missing [lines 207-210]. Additionally, we confirm that other data values are complete, as they are sourced from the Gaia DR3 catalogue.

      • Smoothing Splines: The necessity of using smoothing splines is highlighted to achieve a consistent number of points in the time series and to minimize fluctuations or noise [lines 245-252].

  1. Reviewer Comment 2: The manuscript does not provide a clear rationale for the selection of certain models over others.

  • To address this, we have added a new subsection (4.2) titled "Choosing the Right Neural Network Architecture for Time-Series Data." This section provides a detailed explanation of the rationale behind our model choices and the criteria used to select the appropriate architectures for our time-series data.

  1. Reviewer Comment 3: The manuscript briefly discusses hyperparameter tuning but it does not provide any specifics regarding the search space, optimization techniques or the computational resources that were employed.

  • To address your concerns, we have made the following updates:

    • Optimization Techniques and Search Space: We have added details to subsection 4.1, including information about the search space and the optimization techniques used. Specifically, we discuss the Adam optimizer, the learning rate settings, and the implementation of early stopping.

    • Computational Resources: Information about the computational resources employed is now included in Section 5.1, "Experiment Setup."

We hope these additions provide the necessary specifics and enhance the clarity of our approach.

  1. Reviewer Comment 4: The manuscript does not include confidence intervals or statistical significance testing for metrics such as MAE and R². Kindly include.

  • To address your concerns, we have made the following updates:

    • Standard Deviation: We have added the standard deviation for each metric listed in Table A1 to provide a confidence interval.

    • Open Source Script: An open-source script for calculating the standard deviation has been added to the repository, ensuring transparency and reproducibility. All code is open-source and available within the GitHub repository

    • Additional Information: We have included a new sentence in the "Results of the Experiments" section to discuss the addition of standard deviation and its relevance.

  1. Reviewer Comment 5: The interpretability of the deep learning models is not addressed in the manuscript.

  • The empirical relationship between the shape of the light curve of fundamental-mode (ab type) RR Lyrae stars and their metallicity, as established by Jurcsik and Kovács (1996), has traditionally been analyzed using Fourier decomposition methods. Our work aims to achieve similar results using alternative methods, avoiding the need for Fourier decomposition. To clarify, we have focused on developing and evaluating models based on their performance rather than interpretability. However, we understand the importance of addressing how our models relate to the established methods. If our approach does not fully meet the interpretability expectations, we appreciate further guidance on specific aspects you would like to see addressed.

  1. Reviewer Comment 6: The abstract could be enhanced by including more detailed information regarding the results, such as the precise performance metrics.

  • We have revised the abstract to include more detailed information regarding the results. Specifically, we have added the precise performance metrics, including R^2, RMSE, MAE, wRMSE, and MAE.

  1. Reviewer Comment 7: Publication years of certain important references in the introduction section are absent.

  • We have updated the manuscript to include both the author names and publication years for all relevant references. The first one reference in the introduction section, has been corrected to include the author and year of the publication.

  1. Reviewer Comment 8: Insufficient captions in certain figures and tables.

  • We have improved the captions for the following items to provide clearer and more detailed descriptions:

    • Figures 1, 2, and 4

    • Tables 2, 3, and A1

  1. Reviewer Comment 9: The manuscript contains minor grammatical errors.

  • We have revised the manuscript to correct these errors throughout the manuscipt. These changes have been made in alignment with the suggestions from all the reviewers as well.

  1. Reviewer Commen 10: The manuscript mentions the availability of code, but it does not provide a direct link.

  2. We have included the direct link to the repository at the end of subsection 4.2. For your convenience, here is the repository link: https://github.com/LorenzoMonti/metallicity_rrls. Additionally, we have added the following statement before the acknowledgments section to ensure completeness:

"Data Availability Statement: All code is open-source and available within the GitHub repository. The dataset can be downloaded from the Gaia Archive site, specifically from the Gaia DR3 page, according to the selection criteria presented in Section 3.". We hope this provides the necessary access and information.

Thank you once again for your detailed and constructive feedback. Your suggestions have significantly improved the clarity and quality of our manuscript. We believe that the revisions and additions address your concerns effectively. If you have any further questions or require additional information, please do not hesitate to let us know.

Sincerely,

Lorenzo Monti, PhD

National Institute for Astrophysics – OAS (Bologna)